# Detection of Plasmid-Mediated Resistance against Colistin in Multi-Drug-Resistant Gram-Negative Bacilli Isolated from a Tertiary Hospital

**DOI:** 10.3390/microorganisms11081996

**Published:** 2023-08-03

**Authors:** Mario Galindo-Méndez, Humberto Navarrete-Salazar, Reinaldo Pacheco-Vásquez, Devanhí Quintas-de la Paz, Isabel Baltazar-Jiménez, José David Santiago-Luna, Laura Guadarrama-Monroy

**Affiliations:** 1Laboratorios Galindo SC, Av Juárez 501-A, Col Centro, Oaxaca 68000, Oax, Mexico; 2School of Medicine, Universidad Anáhuac Oaxaca, Blvd. Guadalupe Hinojosa de Murat 1100, San Raymundo Jalpam 71248, Oax, Mexico; 3Centro Médico ISSEMYM Toluca, Av. Baja Velocidad KM. 57.5, Carr. Mex./Tol. Col. San Jeronimo Chicahualco, Metepec 52170, Edomex, Mexico

**Keywords:** antibiotic resistance, *mcr*-1, plasmid-mediated colistin resistance, O25b-ST131, CTX-M

## Abstract

The aim of this study was to determine the prevalence of plasmid-mediated colistin resistance *mcr*-1 to *mcr*-5 genes among colistin and multi-drug-resistant Gram-negative bacilli strains isolated from patients in a tertiary hospital in Toluca, Mexico. The presence of *mcr* genes among the 241 strains collected was assessed by PCR. In the case of *mcr*-carrying *E. coli*, further PCR tests were performed to determine the presence of *bla*_CTX-M_ and whether the strains belonged to the O25b-ST131 clone. Conjugation experiments were also carried out to assess the horizontal transmission of colistin resistance. A total of twelve strains (5.0%), of which four were *E. coli*; four were *P. aeruginosa*; three were *K. pneumoniae,* and one *E. cloacae*, were found to be resistant to colistin. Of these strains, two *E. coli* isolates were found to carry *mcr*-1, and Southern blot hybridization demonstrated its presence on an approximately 60 kb plasmid. Both *mcr*-1-carrying *E. coli* strains were found to co-express *bla*_CTX-M_, belong to the O25b-ST131 clone, and horizontally transmit their colistin resistance. The results of this study confirm the presence of plasmid-mediated colistin resistance in hospitalized patients in Mexico and demonstrated that the multi-drug-resistant O25b-ST131 *E. coli* clone can acquire *mcr* genes and transmit such resistance traits to other bacteria.

## 1. Introduction

The use of antibiotics in humans and animals represents the cornerstone of modern medicine. Since their discovery, antibiotics have revolutionized the treatment of bacterial infections around the world. However, bacteria have evolved mechanisms to become resistant to these agents, and a steady increase in this phenomenon has made antimicrobial resistance one of the major health problems affecting humankind. 

In the case of Gram-negative bacteria, infections caused by multi-drug-resistant *Enterobacterales*, *Pseudomonas aeruginosa,* and *Acinetobacter* sp. are on the rise [1]. This reality has led to the use of old antibiotics, such as colistin, as a last-resort antimicrobial against these pathogens. Colistin, a polypeptide antibiotic also known as polymyxin E, was discovered in 1949 and was largely used in the 1950s; however, due to its neuro- and nephro-toxicity, it was abandoned in the 1980s in favor of other antibiotics [2]. Colistin exerts its antibacterial activity against Gram-negative organisms by electrostatically binding the negatively charged phosphate groups of lipid A in the bacterial lipopolysaccharide (LPS), disrupting the integrity of the bacterial membrane [3].

Currently, antibiotic resistance against colistin is not as worrisome as that against other antimicrobials; however, due to the overuse and misuse of this antibiotic among humans and animals, colistin resistance is on the rise, as described in several reports from different nations [4,5,6]. Resistance against this agent in *Enterobacterales*, *P. aeruginosa,* and *Acinetobacter* sp. was originally described as being chromosomally encoded. This resistance mechanism was identified to be due to the modification of the LPS, through cationic substitutions by the addition of phosphoethanolamine (PEtN) or 4-amino-4-deoxy-L-arabhinose (L-Ara4N), molecules that reduce the net negative charge of the LPS, which impedes the binding of the antibiotic [7]. Specific mutations in any of the chromosomal genes of the two bacterial sensory two-component systems, PhoPQ or PmrAB, or in its negatively regulating gene, *mgrB*, induces the synthesis of the LPS-modifying moieties [7]. However, in 2015, Liu et al. described an additional colistin-resistance mechanism, the first plasmid-mediated resistant gene against this antibiotic, *mcr*-1, in *E. coli* [8]. Shortly after its discovery, *mcr*-1 was identified in over 20 different countries, and, to date, nine additional *mcr*-like genes have been described [9]. Furthermore, *mcr* genes have also been identified in *Pseudomonas aeruginosa* and *Acinetobacter* sp. [10,11]. These plasmid-mediated genes confer bacteria the ability to produce an enzyme, phosphoethanolamine transferase, that catalyzes the addition of substrates that bind the LPS and inhibit the binding of colistin [8].

With the global spread of *mcr* genes among multi-drug-resistant bacterial strains and their further horizontal transmission among bacteria, the effectiveness of colistin is in serious jeopardy. An additional factor that might facilitate the spread of colistin resistance is the likelihood of acquiring this plasmid-mediated resistant trait in multi-drug-resistant *E. coli* clones such as O25b-ST131, the most common multi-drug-resistant high-risk clone associated with extra-intestinal *E. coli* infections around the world [12]. This clone commonly carries resistance genes to antibiotics frequently prescribed in general practice, such as cephalosporins, which are mainly mediated by the *bla*_CTX-M_ gene, as well as quinolones, and can easily be transmitted through the consumption of food [13]. These antibiotics can co-select for colistin-resistant strains and, thus, contribute to the spread of resistance against this agent. ST131 *E. coli* clones carrying colistin resistance genes have already been reported in environmental and clinical samples [14,15].

As the surveillance of antibiotic resistance and research is a key component of the global action plan against antimicrobial resistance [16], the purpose of the current study was to determine the prevalence of the plasmid-mediated colistin resistance genes *mcr*-1 to *mcr*-5 among colistin and multi-drug-resistant *Enterobacterales*, *Pseudomonas aeruginosa* and *Acinetobacter* sp. strains isolated from patients in a tertiary hospital in the city of Toluca, Mexico. Additionally, we sought to assess whether the plasmid-mediated colistin-resistant *E. coli* strains isolated from these patients belonged to the ST131 clone and co-expressed *bla*_CTX-M_ genes.

## 2. Materials and Methods

This was a prospective study conducted between May and October 2022 in coordination with the Microbiology laboratory from Centro Médico ISSEMYM, Toluca, Mexico. Strains included in this research were isolated from cultures obtained by the clinical laboratory of the hospital as part of routine care for hospitalized patients, as instructed by their physician. No additional specimens were obtained for the purposes of this study and no personal information was obtained from patients; therefore, informed consent was not required.

### 2.1. Bacterial Strains, Culture, Identification, and Microbial Susceptibility Testing

Biological samples were plated on blood agar and MacConkey agars and cultured at 35 ± 2 °C for 18 h. Strains growing on the latter medium were identified using the VITEK Compact System (bioMerieux, Marcy l’E’toile, France), following the manufacturer’s instructions. Only strains that were unequivocally identified by the Vitek system to belong to the order *Enterobacterales* or the genus *Pseudomonas aeruginosa* or *Acinetobacter* spp. were included in the study (identification probability > 90%).

The minimum inhibitory concentrations (MIC) of different antibiotics against the strains included in the study were determined using the VITEK Compact System (bioMerieux, Marcy l’E’toile, France) and compared to the Clinical and Laboratory Standard Institute (CLSI) guidelines [17]. Only strains that fulfilled the multi-drug-resistance (MDR), extensively drug-resistant (XDR), or pan-drug-resistant criteria (PDR) established by Magiorakos et al. [18] were included for further testing. MDR, XDR, and PDR strains were initially tested for colistin resistance by the colistin broth disk elution test described by Simner et al. [19]. The MIC of the strains determined to be resistant by this method (≥4 µg/mL) were further measured in duplicate using the micro broth dilution method according to CLSI guidelines [17] and only those confirmed to be colistin-resistant (MIC ≥ 4 µg/mL) were included in the study.

### 2.2. PCR Amplification

DNA was extracted using the boil lysis method, and extracted DNA was tested via PCR for *mcr*-1 to *mcr*-5 genes using the primers described in Table 1. Amplified PCR products were sequenced in both directions and nucleotide sequences were compared against the National Center for Biotechnology Information BLAST database [20]. In the case of colistin-resistant *E. coli* strains carrying *mcr* genes, the presence of the O25b-ST131 clone and *bla*_CTX-M_ genes was determined using previously described PCR primers (Table 1).

### 2.3. Conjugation Experiments

Conjugation experiments were carried out using *E. coli* J53, a sodium azide-resistant strain, as the recipient organism, following the broth-mating method. Briefly, donor and recipient strains were cultured on Luria–Bertani (LB) broth with shaking at 125 rpm at 37 °C to logarithmic phase (OD_600_ = 0.5). Conjugation was performed by mixing donor and recipient strains in a 1:1 ratio in 4 mL of LB broth and incubated overnight at 37 °C without shaking. Transconjugants were selected on Mueller Hinton agar plates containing 100 mg/mL sodium azide and 4 µg/mL of colistin. The presence of *mcr* genes in transconjugants was assessed via antimicrobial susceptibility testing using the micro broth dilution method, as suggested by CLSI [17], as well as PCR and sequencing. The transmission of the *bla*_CTX-M_ gene to transconjugants was confirmed by PCR, and its ESBL phenotype, by the CLSI confirmatory method [17].

### 2.4. S1-Pulsed-Field Gel Electrophoresis (PFGE) and Southern Blot Analysis

To determine the location of transmissible elements, genomic DNA of both *mcr*-1-carrying *E. coli* transconjugant strains was digested with S1 nuclease (Thermo Fisher Scientific, Waltham, MA, USA) and electrophoresed on a CHEF-mapper XA pulsed-field gel electrophoresis system (Bio-Rad Laboratories Inc., Hercules, CA, USA) for 22 h at 14 °C with run conditions of 6 V/cm and pulse times from 2.16 s to 63.8 s. The DNA fragments were transferred to a positively charged nylon membrane (EMD Millipore, Burlington, MA, USA) and then hybridized with an approximately 600 bp digoxigenin-labeled *mcr*-1 probe, according to the manufacturer’s instructions (Roche, Mannheim, Germany). The fragments were then detected using an NBT/BCIP color detection kit (Hoffman-La Roche Ltd., Basel, Switzerland). The *Salmonella enterica* serotype Braenderup H9812 was used as the size marker.

For all experiments, a previously identified *E.coli* strain isolated from a swine farm, resistant to the colistin-carrying *mcr*-1 gene, was included as a positive control [25].

## 3. Results

### 3.1. Colistin Resistance

In total, 241 isolates of MDR, XDR, and PDR strains of *Enterobacterales*, *P. aeruginosa,* and *Acinetobacter* sp. were collected at the clinical laboratory from Centro Médico ISSEMYM Toluca and included in this study. The number of isolates per bacterial species identified is shown in Figure 1.

Among the 241 multi-drug-resistant strains included in the study, 12 (5.0%) were found by the micro-broth dilution method to be resistant against colistin (MIC ≥ 4 µg/mL) and included *E. coli* (*n* = 4), *P. aeruginosa* (*n* = 4), *K. pneumoniae* (*n* = 3), and *E. cloacae* (*n* = 1). The colistin MIC of these 12 strains and their isolation sites are shown in Table 2.

The antibiotic resistance profiles of the 12 strains found to be resistant to colistin against commonly used antibiotics and their respective classification as MDR, XDR, or PDR are depicted in Table 3.

### 3.2. mcr Prevalence

Of all 12 strains resistant to colistin, the multiplex PCR protocol amplified one fragment of approximately 320 bp in two *E. coli* isolates (2207 and 5891) and in the control strain, suggesting the presence of the *mcr*-1 gene (Figure 2). Sequencing in both directions of these amplicons confirmed that both strains carry this gene. PCR targeting was negative on all strains for the *mcr*-2 to -5 genes. Both *mcr*-1-carrying *E.coli* strains were shown to be ESBL producers carrying the *bla*_CTX-M_ gene and belonging to the O25b-ST131 clone.

### 3.3. Conjugation Experiments

Conjugation experiments were conducted on strains 2207 and 5891, with both isolates transmitting their colistin resistance trait to the recipient J53 *E. coli* strain, suggesting that *mcr-1* was located on transferable plasmids. Both transconjugants were shown to be PCR-positive for the *mcr*-1 gene (Figure 2) and sequencing in both directions of these transconjugant amplicons confirmed the horizontal transmission of *mcr*-1. The two parental strains also transmitted their *bla*_CTX-M_ gene, and the recipient strains were confirmed to be ESBL producers by the CLSI confirmatory test [17] as well as de novo resistant to colistin, as assessed by the micro-broth dilution method (≥4 µg/mL). The strain 2207 transconjugant presented an identical colistin MIC to that of the parental strain, while the 5891 transconjugant presented a one to two-fold higher MIC.

### 3.4. Southern Blot Analysis

DNA digestion with S1 nuclease and Southern blot analysis were performed for both transconjugants. The results of these analyses indicated that the *mcr*-1 genes isolated from the two transconjugants were located on a plasmid of approximately 60 kb (Figure 3).

## 4. Discussion

Due to the increasing rates of resistance among Gram-negative bacilli against commonly used antibiotics, mainly in bacterial strains isolated from hospitalized patients, colistin has emerged as one of the last-resort antimicrobials in the treatment of these infections. Unfortunately, resistance against this agent is on the rise, particularly in Asia, which is mainly attributed to its overuse in veterinary medicine [26].

One of the main objectives of the current study was to determine the prevalence of colistin resistance among multi-drug-resistant *Enterobacterales*, *P. aeruginosa,* and *Acinetobacter* sp. strains isolated from patients in a tertiary hospital in Mexico. In total, 241 strains were included in the study, of which 12 (5.0%) were found to be resistant to colistin according to the CLSI guidelines. Of the 12 colistin-resistant strains, 4 were *P. aeruginosa*, with a colistin-resistance prevalence of 7.5%, (4/53); 4 were *E. coli*, with a prevalence of 2.5% (4/162); 3 were *K. pneumoniae*, with a prevalence of 15.0% (3/20), and 1 was *E. cloacae*, with a prevalence of 50.0% (1/2). Although *K. pneumoniae* presented one of the highest prevalence levels of colistin resistance, no PDR strains were found, unlike *P. aeruginosa*, for which one strain was found to be resistant to all tested antibiotics. In this study, no *Acinetobacter* sp. isolates were found to be colistin-resistant.

The antibiotic resistance profiles of the colistin-resistant strains found in this study are shown in Table 3. As all 241 strains included in the current study were multi-drug-resistant isolates, the antibiotic resistance among them was high. In the case of *P. aeruginosa*, colistin-resistant strains presented an extremely high prevalence of resistance against carbapenems (100%), ceftazidime (100%), ciprofloxacin (100%), and amikacin (100%) and low resistance to tigecycline (25%). For *Enterobacterales*, higher rates of resistance were found against ciprofloxacin (100.0%), trimethoprim/sulfamethoxazole (87.5%), cephalosporines (62.5%), and ceftazidime (62.0%); no resistance was found against amikacin. Finally, resistance against at least one of the carbapenems was found in one *E. coli* (1/4) and one *K. pneumoniae* (1/3) strain.

The prevalence of colistin resistance in this study is in agreement with reports in different geographical areas of the world among muti-drug-resistant strains isolated from hospitalized patients [10,27,28], but higher than the 1.26% worldwide prevalence reported by Dadashi et al. [29]. In Mexico, reports on the prevalence of colistin-resistant strains are scarce; however, in a recent report by the Red Temática de Investigación y Vigilancia de la Farmacorresistencia (INVIFAR network), that included clinical strains isolated from different parts of the country, the colistin resistance prevalence among *K. pneumoniae* strains was 17.8% [30], a rate similar to that found in the current study among isolates of this bacterial species (3/20 = 15.0%). Unlike the INVIFAR report, where no resistance to colistin was found among *E. coli* strains, we found that 2.5% of the *E. coli* strains included in the current study presented this resistance trait. These results confirm that colistin resistance among clinical isolates in our country is already a reality and should be carefully monitored. 

As the world has been witnessing an increase in resistance against colistin, this phenomenon has been mainly attributed to the emergence and dissemination of plasmid-mediated (*mcr*-1 to *mcr*-10) genes among bacteria. The first *mcr* gene (*mcr*-1) was originally described in 2015 in China [8], and in Mexico, *mcr*-1 was first identified in 2019 in an *E. coli* strain isolated from a fecal sample from a child [31]. The presence of this plasmid-mediated gene in our country was recently confirmed in *K. pneumoniae* strains isolated from different clinical samples [30]. In the present study, only two *E. coli* strains, both isolated from female patients with urinary tract infections that required hospitalization, were found to carry the *mcr*-1 gene and both strains were able to efficiently transfer this colistin-resistant determinant to recipient strains by means of an approximately 60 kb plasmid, as demonstrated by Southern blot analysis. No other *mcr* gene was found in the strains analyzed and no other bacterial species in addition to *E. coli* were found to carry *mcr*-mediated colistin resistance genes. In this study, *mcr* genes were not found in *P. aeruginosa*. However, as colistin is the last-resort antibiotic against strains of this organism resistant to carbapenems, reports of *mcr* genes among multi-drug-resistant strains of this species have started to emerge [32,33], and the horizontal transmission of *mcr*-1 genes has been shown to occur from *P. aeruginosa* to other bacterial species [34], the surveillance of *mcr*-carrying *P. aeruginosa* strains remains highly encouraged.

The low prevalence of *mcr* genes found in this investigation among colistin-resistant strains (2/12) suggests that in our population, resistance to this agent might be mainly driven, as suggested by Gogry et al. [26], by chromosomal mutations, such as those in the genes regulating the two-component systems, PhoPQ or PmrAB, or in its negatively regulating gene, *mgrB*, by *mcr* genes different from the *mcr*-1 to *mcr*-5 genes included in this investigation or by unknown colistin resistance mechanisms. The results of the current investigation are in agreement with those found in a recent report by Garza-Ramos et al. who found that the more prevalent colistin-resistant mechanism in Mexico is not driven by *mcr* genes but by *mgrB* mutations [30]. Moreover, the low prevalence of *mcr* genes in our population might be partially explained by the fact that colistin-resistance in *Enterobacterales* has been linked mainly to *mgrB* mutations [35], rather than to plasmid-mediated genes. In the case of *P. aeruginosa*, the main drivers of colistin resistance have also been shown to be mainly due to chromosome mutations leading to the overexpression of efflux pumps or defective biofilm formation [36].

The relatively low prevalence of *mcr* genes among colistin-resistant Gram-negative bacteria found in this study (16.7%) is higher than the values reported for *E.coli* in Egypt (7.5%) [37] and similar to those described in Ecuador (20.0%) [38] but lower than the prevalence shown in a study in Nepal [39] and Peru [40] where the prevalence of strains that serve as carriers of *mcr* genes were reported to be high among colistin-resistant *E. coli* and *K. pneumoniae*. The results of the current research confirm the fact that the horizontal transmission of colistin resistance has spread at different rates among different geographical areas of the world. It is also important to note that in this study, *mcr* genes were searched for in colistin-resistant strains and only multi-drug-resistant isolates were included, but different studies have shown that colistin-susceptible *Enterobacterales* can carry *mcr* genes [41,42]. Thus, the prevalence of *mcr*-carrying strains at the Microbiology laboratory from Centro Médico ISSEMYM, Toluca, Mexico, could be higher than the levels indicated by the results in this study. Further research is needed to understand the epidemiology of the additional mechanisms of colistin resistance to those identified in the current study. 

Several reports have shown that Gram-negative bacilli can co-harbor *mcr* genes and other plasmid-mediated antibiotic resistance traits such as those encoding for carbapenemase and ESBL production. Since colistin is mainly indicated as a last-resort antibiotic, the co-expression of colistin and carbapenemase resistance is worrisome among the medical community and, understandably, has been more thoroughly studied and more commonly demonstrated among *Enterobacterales* [30,43] than its co-expression with extended-spectrum-β-lactamases. However, bacterial strains co-harboring *mcr* and ESBL genes should also be carefully monitored. In addition, to encode for intrinsic resistance against cephalosporines, penicillins, and monobactams, plasmids carrying ESBL-encoding genes can also harbor resistance genes against other commonly used antibiotics such as ciprofloxacin, trimethoprim/sulfamethoxazole, and aminoglycosides [44]. When administered, any of these antibiotics could co-select for colistin-resistant strains and contribute to the spread of resistance against this antimicrobial. In the present study, the two *mcr*-1-carrying *E.coli* strains were additionally found to carry *bla*_CTX-M_ genes, a phenomenon that has also been demonstrated in human isolates in Qatar [45], Peru [46], and the Indian Ocean Commission [47], showing that the co-expression of plasmid-mediated resistance against colistin and ESBL production has spread to different regions of the world. 

In the current study, both *mcr*-1- and CTXM-carrying *E. coli* strains were able to horizontally transmit their colistin resistance and ESBL production to the recipient strain, suggesting that conjugation may play a role in the spread of colistin resistance. In this study, only a minority of colistin-resistant isolates were found to carry *mcr* genes and in the two *mcr*-1-carrying strains, no carbapenem resistance was detected, leaving other therapeutic options available for the treatment of infections caused by these strains. However, as different studies have shown that the horizontal transmission of *mcr*-1 genes occurs in food [48] and animals [49], this mechanism might be responsible for the steady increase in colistin resistance among Gram-negative bacilli around the globe. 

The highly antibiotic-resistant clone ST131, predominantly serogroup O25b, is considered the dominant extra-intestinal pathogenic *E. coli* around the world, as well as a frequent cause of urinary tract infections [12,13]. In the current study, the two *E. coli* strains carrying the *mcr*-1 and *bla*_CTX-M_ genes were isolated from patients with urinary tract infections and were resistant to most classes of antibiotics, demonstrating sensitivity only to the carbapenems and, in the case of one strain (2207), to cefepime. In addition, these two strains were found to belong to the ST131-O25b clone, an *E. coli* clone that commonly exhibits resistance to quinolones, trimethoprim/sulfamethoxazole, and aminoglycosides, and is recognized as the primary lineage responsible for the spread of *bla*_CTX-M_ genes [50]. The results of this study support previous findings on strains isolated from humans [15,51], showing that the already highly resistant clone ST131 can acquire plasmid-mediated colistin resistance genes. As this clone can be transmitted from person to person and through the consumption of contaminated food [13], and given that its prevalence in the feces of healthy humans is on the rise [52], the spread of this multi-drug-resistant clone could be facilitated by a lack of hygiene, which suggests that less privileged areas of the world might see an increase in the prevalence of this clone. If *mcr*-1-carrying O25b-ST131 *E. coli* clones expand to different geographical areas, commonly used antibiotics, such as ciprofloxacin, cephalosporines, and trimethoprim/sulfamethoxazole, could co-select for colistin-resistant strains and contribute to the spread of resistance against this last-resort antibiotic.

Limitations of the current study include its small sample size, unicentric design, and short timeframe. In addition, as colistin resistance can be mediated by chromosomal mutations, or by plasmid-mediated *mcr* genes different from those included in the current study, it cannot be concluded that the colistin resistance found in the two *E. coli* strains reported herein might be mediated by the *mcr*-1 genes alone or that such resistance is a combination of additional resistance mechanisms. Lastly, not all known *mcr* variants were analyzed, thus, the actual plasmid-mediated colistin resistance prevalence at our institution might be higher than the levels suggested by these results.

In order to further understand the epidemiology, dissemination, and evolution of colistin resistance in our country, larger, multicentric studies must be performed. In addition, as *mcr*-mediated resistance against this agent is on the rise, these studies must include further analysis of the plasmid types involved in the horizontal transmission of *mcr* genes, as well as identify whether other mobile genetic elements, such as transposons or integrons, are involved in the spread of this worrisome antibiotic resistant trait. This information can help the medical and scientific community and policymakers to understand how these antibiotic-resistant traits move among bacteria and whether interventions can be applied to stop their spread. 

In conclusion, this study, albeit small, demonstrated that the emergence and spread of *mcr*-carrying strains among humans is a reality in Mexico. In addition, the presence of this resistant trait among highly resistant and easily transmissible O25b-ST131 *E. coli* clones further complicates the antibiotic resistance scenario in our country. Further surveillance studies are needed in other hospitals and among other ambulatory patients in Mexico to determine the magnitude of the problem of colistin resistance, especially that mediated by *mcr* genes, in order to establish policies aimed at optimizing antibiotic stewardship programs to reduce the dissemination of resistance against this last-resort antibiotic.

## Figures and Tables

**Figure 1 microorganisms-11-01996-f001:**
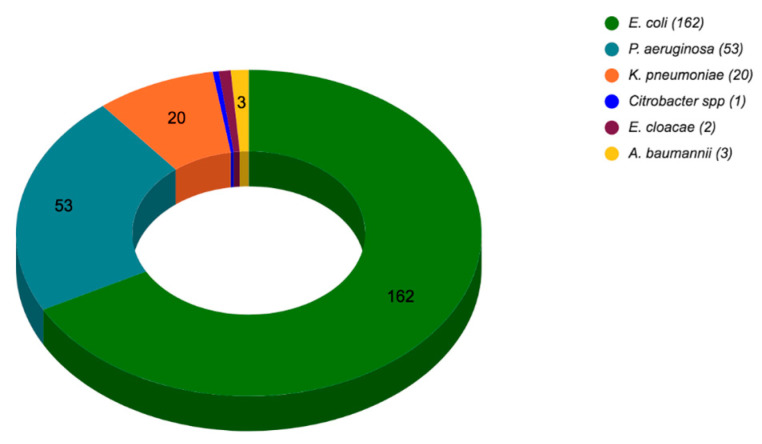
The number of strains isolated per bacterial species.

**Figure 2 microorganisms-11-01996-f002:**
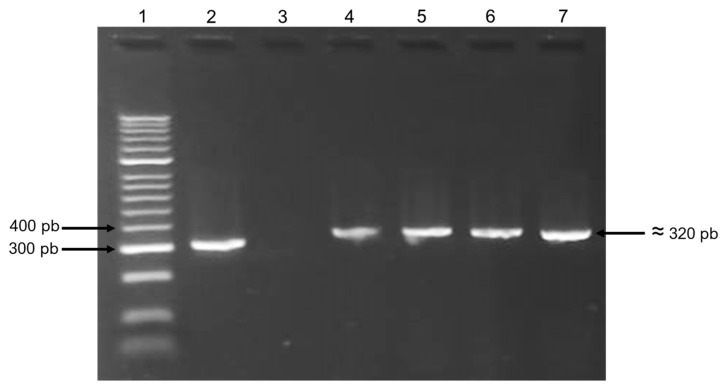
PCR amplification results for the *mcr*-1 gene. Lane 1: molecular marker; lane 2: *mcr*-1 positive control; lane 3: *mcr*-1 negative control; lane 4: *E. coli* strain 5891; lane 5: *E. coli* conjugate strain 5891; lane 6: *E. coli* strain 2207; and lane 7: *E. coli* conjugate strain 2207.

**Figure 3 microorganisms-11-01996-f003:**
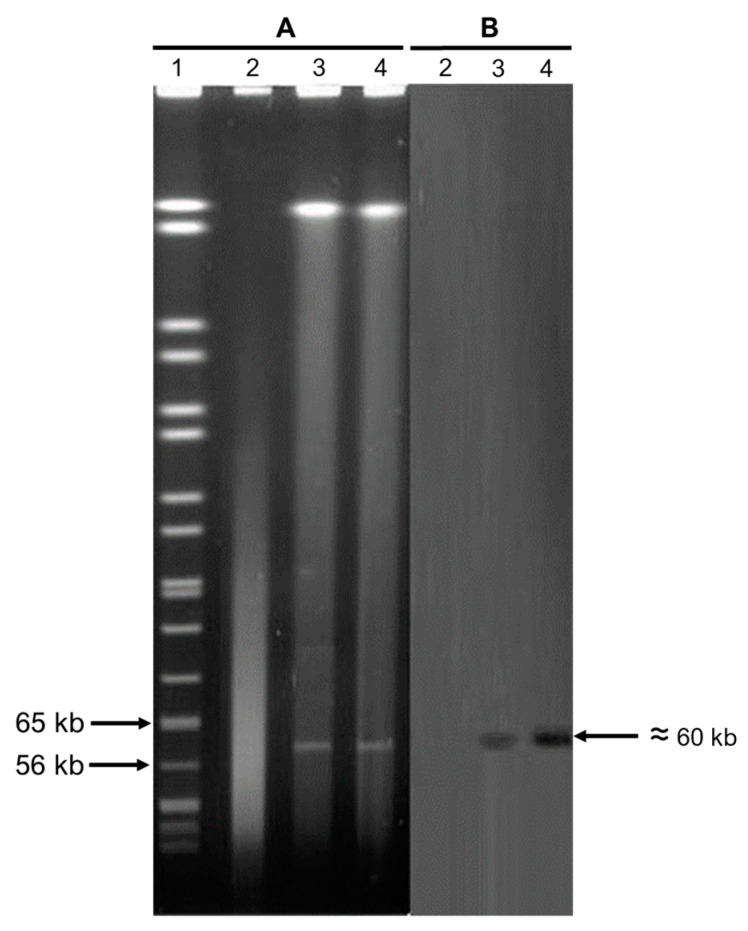
Southern blot analysis with a probe for *mcr-1*. (A) S1-nuclease-PFGE of *E.coli* transconjugants. (B) Southern blot hybridization with *mcr*-1 probe. Lane 1: marker; lane 2: *E. coli* J53 strain; lane 3: *E. coli* strain 2207 transconjugant; and lane 4: *E. coli* strain 5891 transconjugant.

**Table 1 microorganisms-11-01996-t001:** Primers and PCR conditions used to determine the presence of *mcr* and *bla*_CTX-M_ genes and O25b-ST131 clones.

Amplified Gene	Primers Sequence (5′–3′)	Reference
*mcr*-1-fw	AGTCCGTTTGTTCTTGTGGC	[21]
*mcr*-1-rv	AGATCCTTGGTCTCGGCTTG	
*mcr*-2-fw	CAAGTGTGTTGGTCGCAGTT	[21]
*mcr*-2-rv	TCTAGCCCGACAAGCATACC	
*mcr*-3-fw	AAATAAAAATTGTTCCGCTTATG	[21]
*mcr*-3-rv	AATGGAGATCCCCGTTTTT	
*mcr*-4-fw	TCACTTTCATCACTGCGTTG	[21]
*mcr*-4-rv	TTGGTCCATGACTACCAATG	
*mcr*-5-fw	ATGCGGTTGTCTGCATTTATC	[22]
*mcr*-5-rv	TCATTGTGGTTGTCCTTTTCTG	
pabB-fw	TCCAGCAGGTGCTGGATCGT	[23]
pabB-rv	GCGAAATTTTTCGCCGTACTGT	
*bla*_CTX-M_-fw	TTTGCGATGTGCAGTACCAGTA	[24]
*bla*_CTX-M_-rv	CGATATCGTTGGTGGTGCCATA	

**Table 2 microorganisms-11-01996-t002:** Colistin MIC of strains identified as resistant and their isolation sites.

Strain	Micoorganism	Colistin MIC (µg/mL)	Isolation Site
744	*P. aeruginosa*	8	Blood
1308	*K. pneumoniae*	16	Respiratory secretions
2207	*E. coli*	4	Respiratory secretions
2230	*E. coli*	4	Urine
2445	*E. coli*	8	Renal abscess
2892	*P. aeruginosa*	16	Respiratory secretions
3148	*P. aeruginosa*	8	Urine
3172	*K. pneumoniae*	4	Blood
3196	*P. aeruginosa*	4	Urine
3202	*K. pneumoniae*	4	Respiratory secretions
3271	*E. cloacae*	16	Respiratory secretions
5891	*E. coli*	4	Urine

**Table 3 microorganisms-11-01996-t003:** Antibiotic resistance profile of colistin-resistant strains.

Colistin Resistant Bacteria	Antibiotic	Antibiotic Resistance Prevalence	Acquired Resistance Profile
*P. aeruginosa* (*n* = 4)	Ceftazidime	4/4 (100%)	MDR: 0
	Cefepime	4/4 (100%)	XDR: 3
	Amikacin	4/4 (100%)	PDR: 1
	Ciprofloxacin	4/4 (100%)	
	Piperacillin/tazobactam	2/4 (50.0%)	
	Imipenem	4/4 (100%)	
	Ceftazidime	4/4 (100%)	
	Meropenem	4/4 (100%)	
	Gentamicin	4/4 (100%)	
	Tigecycline	1/4 (25%)	
*E. coli* (*n* = 4)	Ampicillin/sulbactam	3/4 (75%)	MDR: 2
	Cefuroxime	3/4 (75%)	XDR: 2
	Cefotaxime	3/4 (75%)	PDR: 0
	Ceftazidime	3/4 (75%)	
	Ceftriaxone	3/4 (75%)	
	Cefepime	2/4 (50%)	
	Ertapenem	1/4 (25%)	
	Meropenem	0/4 (0%)	
	Amikacin	0/4 (0%)	
	Gentamicin	2/4 (50%)	
	Ciprofloxacin	4/4 (100%)	
	Trimethoprim/sulfamethoxazole	4/4 (100%)	
*K. pneumoniae* (*n* = 3)	Ampicillin/sulbactam	3/3 (100%)	MDR: 1
	Cefuroxime	2/3 (66.7%)	XDR: 2
	Cefotaxime	2/3 (66.7%)	PDR: 0
	Ceftazidime	2/3 (66.7%)	
	Ceftriaxone	2/3 (66.7%)	
	Cefepime	2/3 (66.7%)	
	Ertapenem	1/3 (33.3%)	
	Meropenem	0/3 (0%)	
	Amikacin	0/3 (0%)	
	Gentamicin	2/3 (66.7%)	
	Ciprofloxacin	3/3 (100%)	
	Trimethoprim/sulfamethoxazole	3/3 (100%)	
*E. cloacae* (*n* = 1)	Cefuroxime	1/1 (100%)	MDR: 1
	Cefotaxime	0/1 (0%)	XDR: 0
	Ceftazidime	0/1 (0%)	PDR: 0
	Ceftriaxone	0/1 (0%)	
	Cefepime	0/1 (0%)	
	Ertapenem	0/1 (0%)	
	Meropenem	0/1 (0%)	
	Amikacin	0/1 (0%)	
	Gentamicin	1/1 (100%)	
	Ciprofloxacin	1/1 (100%)	
	Trimethoprim/sulfamethoxazole	0/1 (100%)	

## Data Availability

Not applicable.

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
