# Peer review of "Detection of Plasmid-Mediated Resistance against Colistin in Multi-Drug-Resistant Gram-Negative Bacilli Isolated from a Tertiary Hospital"

_microorganisms, 2023, doi:10.3390/microorganisms11081996_

Round 1
Reviewer 1 Report
This manuscript determined the prevalence of the plasmid-mediated colistin resistance genes mcr -1 to mcr -5 among colistin and multi-drug resistant strains isolated from local patients The authors also assessed whether plasmid-mediated colistin-resistant E. coli strains isolated from these patients belonged to the ST131 clone and co-expressed blaCTX-M genes, validated by conjugation experiments. This manuscript matches the scope of the journal and showed logical structure. Several recommendations have been suggested to improve this manuscript.
This manuscript applied culture based method according to the guidelines of CLSI to isolate drug resistant bacteria phenotypes. This method is a good starting point but can not identify whether the resistance was mediated by the colistin resistance genes or by other multidrug resistance mechanisms. The result that only 2 out of 12 phenotypic colistin-resistant isolates were carrying colistin resistance gene also suggested that the colistin resistance defined in this study is a bit ambiguous. I recommend the authors to discuss and better define the difference between the colistin resistance phenotype and genotype of the strains studied in this manuscript, with a highlight on genotype resistance since they might go mobile.
Another reminder is that the conjugation experimental design should be given in more detail (cell density, conjugation experiment duration, media composition, biological replicate numbers, etc.). In addition, the current selection method of transconjugant can not ensure the transfer was mediated by plasmids. Other mobile genetic elements such as transposons and integrons might also mediate the mcr-1 gene transfer. It is suggested to use PCR or sequencing approach to confirm the presence and transfer of plasmids.
Overall, I think this manuscript is of scientific interest. But the conclusion of the presence and transfer plasmid-mediated colistin resistance genes need to be made based on more solid evidence.
The quality of English language is good for this manuscript.
Reviewer 2 Report
The objective of this article is to investigate the prevalence of colistin resistance in strains of multidrug-resistant Escherichia coli, Pseudomonas aeruginosa, and Acinetobacter isolated from patients at a tertiary hospital in Mexico. Additionally, the authors aim to evaluate whether the colistin resistant strains isolated from these patients are caused by cytoplasmic mediated resistance gene mcr transmission. The conclusions derived here are interesting. I still have some questions that need to be addressed.
1. The article reported the inclusion of 241 multidrug-resistant strains, including MDR, XDR, and PDR types, from clinical laboratories. However, it did not specify the standards and methods used for bacterial genus isolation and identification. Additionally, accompanying images and results of morphology, staining, or biochemical testing should be provided with the findings.
2. The inclusion of multiplex PCR amplification results for the mcr-1 gene would enhance the robustness of the findings; however, these were not observed in result 3.2.
3. The low prevalence of the MCR gene among the identified colistin-resistant strains mentioned in the article suggests that chromosomal mutations or plasmids carrying other genes are primarily responsible for colistin resistance in this population, which was not included in this study. Therefore, it is expected to expand the research scope further in order to investigate the primary mechanisms of colistin resistance.
Round 2
Reviewer 1 Report
The authors responded to my comments appropriately and updated the manuscript accordingly. I understand due to budget and time limitations that additional experiments might not be able to include in the revision. Still, I recommend the authors consider my suggestions for additional scientific evidence to support the claims of this manuscript.
English language of this manuscript is satisfiable.
Reviewer 2 Report
No comment